# Maternal Metformin Treatment during Gestation and Lactation Improves Skeletal Muscle Development in Offspring of Rat Dams Fed High-Fat Diet

**DOI:** 10.3390/nu13103417

**Published:** 2021-09-28

**Authors:** Jiaqi Cui, Lin Song, Rui Wang, Shuyuan Hu, Zhao Yang, Zengtie Zhang, Bo Sun, Wei Cui

**Affiliations:** 1Department of Endocrinology and Second Department of Geriatrics, The First Affiliated Hospital of Xi’an Jiaotong University, Xi’an 710061, China; cuijiaqi9@163.com; 2Department of Physiology and Pathophysiology, School of Basic Medical Sciences, Xi’an Jiaotong University Health Science Center, Xi’an 710061, China; lsong1030@xjtu.edu.cn (L.S.); 160020@peihua.edu.cn (R.W.); hushuyuan0113@stu.xjtu.edu.cn (S.H.); 3Key Laboratory of Environment and Genes Related to Diseases, Ministry of Education of China, Xi’an Jiaotong University, Xi’an 710061, China; 4Department of Obstetrics and Gynecology, The First Affiliated Hospital of Xi’an Jiaotong University, Xi’an 710061, China; yangzhao123@stu.xjtu.edu.cn; 5Department of Pathology, Xi’an Jiao Tong University Health Science Center, Xi’an 710061, China; zengtiez@163.com

**Keywords:** maternal diet, metformin, skeletal muscle, mitochondria, AMPK, mTOR

## Abstract

Maternal high-fat (HF) diet is associated with offspring metabolic disorder. This study intended to determine whether maternal metformin (MT) administration during gestation and lactation prevents the effect of maternal HF diet on offspring’s skeletal muscle (SM) development and metabolism. Pregnant Sprague-Dawley rats were divided into four groups according to maternal diet {CHOW (11.8% fat) or HF (60% fat)} and MT administration {control (CT) or MT (300 mg/kg/day)} during gestation and lactation: CH-CT, CH-MT, HF-CT, HF-MT. All offspring were weaned on CHOW diet. SM was collected at weaning and 18 weeks in offspring. Maternal metformin reduced plasma insulin, leptin, triglyceride and cholesterol levels in male and female offspring. Maternal metformin increased MyoD expression but decreased Ppargc1a, Drp1 and Mfn2 expression in SM of adult male and female offspring. Decreased MRF4 expression in SM, muscle dysfunction and mitochondrial vacuolization were observed in weaned HF-CT males, while maternal metformin normalized them. Maternal metformin increased AMPK phosphorylation and decreased 4E-BP1 phosphorylation in SM of male and female offspring. Our data demonstrate that maternal metformin during gestation and lactation can potentially overcome the negative effects of perinatal exposure to HF diet in offspring, by altering their myogenesis, mitochondrial biogenesis and dynamics through AMPK/mTOR pathways in SM.

## 1. Introduction

Maternal nutrition status is an important *intro utero* factor that affects fetal growth and development. Several studies have suggested that high-fat (HF) diets during gestation and lactation may lead to the development of obesity and insulin resistance in adulthood, and affect the structural and functional development of skeletal muscle (SM) in offspring [1,2].

SM is a potential anti-obesity target because it comprises approximately 40% of human body mass, and is the chief peripheral tissue responsive to insulin-stimulated uptake of glucose and fatty acids [3,4]. The embryonic period plays a determinant role in the development of the muscle mass during the life course because the number of muscle fibers is set at birth [5]. Postnatal muscle growth (hypertrophy) is characterized by an increase in the size of existing muscle fibers [6]. Myogenic regulatory factor family (MRFs) including Myf5, MyoD, Myogenin (Myog) and MRF4 control muscle tissue differentiation during development [7].

Mitochondria, as the central link of energy metabolism, are the energy factories of SM cells [8]. As a central regulator of mitochondrial function, peroxisome proliferator-activated receptor gamma coactivator 1 alpha (Ppargc1a) plays an important role in mitochondrial biogenesis [9]. Mitochondrial transcription factor A (Tfam) activates mitochondrial DNA (mtDNA) transcription, and protects mtDNA from mutation by creating a mitochondrial nucleoid structure around it [10]. In response to proteasome dysfunction, the transcriptional factor NF-E2-related factor 1 (Nrf1) travels to the nucleus to activate the genes that code for proteasomes [11]. Recent literature has revealed that mitochondrial structure is regulated by the dynamics of its membrane fusion and fission [12]. In mammals, mitochondrial dynamics are regulated by some GTPase enzymes: mitochondrial fusion is mediated by mitofusin 1 (Mfn1), Mfn2 and optic atrophy 1 (Opa1), in the meantime, mitochondrial fission is mainly controlled by GTPase dynamin-related protein 1 (Drp1) [13]. 

Metformin is the first choice glucose-lowering agent in patients with type 2 diabetes (T2D) and obesity [14]. Patients with gestational diabetes mellitus (GDM) have been traditionally recommended a diet and insulin when required [15]. Although there are enough data to show that metformin is dependable and effective for women with GDM, there are limited data on the long-term effects of maternal metformin administration on offspring [16]. Recent literature has proposed that metformin inhibits glucose production and ameliorates insulin resistance, mainly through mediating AMP-activated protein kinase (AMPK) [17]. Meanwhile, the activation of AMPK may contribute to the increase in SM glucose utilization [18] and improve mitochondrial respiratory activity [19]. The size, shape, location and quantity of mitochondria can change with the fluctuation of energy supply and demand [20]. The cellular energy sensor, mammalian target of rapamycin (mTOR), not only controls energy homeostasis by means of the transcriptional control of mitochondrial oxidative function [21], but also is a widely recognized controller of muscle mass [22]. Therefore, metformin can inhibit mTORC1 signaling through dose-dependent mechanisms relating to AMPK [23].

Although metformin is not the first-line treatment for GDM as it can pass through the placental barrier [24], a cohort study suggested that there was no difference in the evaluation of growth and development between children of metformin-treated mothers and insulin-treated mothers [25]. There are several studies reporting effects of metformin administration to obese mothers during gestation and lactation on liver and subcutaneous adipose [26], brain [27] and brown adipose tissue [28] in the offspring, while there are few studies on the effect of maternal metformin on offspring’s SM. We tried to determine whether maternal metformin administration during gestation and lactation can improve the adverse effects of maternal HF diet on the SM development and metabolism of offspring in this study.

## 2. Materials and Methods

### 2.1. Animals

All animal experiments strictly comply with the national regulations on the administration of experimental animals, and have been approved by the ethics committee of our institution. Twenty-seven virgin female Sprague-Dawley rats (240–250 g) were purchased from the Experimental Animal Center of Xi’an Jiaotong University.

Rats were individually housed in a temperature- and light-controlled room, and fed a standard laboratory chow diet (Beijing Ke Ao Xie Li, Beijing, China). Rats had free access to food and water at any time. After a one-week habituation, female rats were mated with male rats (280∼300 g) for breeding purposes. Copulatory plug formation was used as confirmation of mating. After mating, female rats were randomized to receive either a control diet (CHOW; Beijing Ke Ao Xie Li, Beijing, China; n = 13) or a high-fat diet (HF; Research Diets D12492, New Brunswick, NJ, USA; n = 14). The CHOW diet had a caloric composition of 65.1% carbohydrate, 23.1% protein and 11.8% fat, whereas the HF diet had a composition of 20% carbohydrate, 20% protein and 60% fat. All rats received their respective diets throughout gestation and lactation. The CHOW-fed rats and HF-fed rats were further separated into two subgroups, with the metformin group (CH-MT, n = 5 and HF-MT, n = 6) receiving metformin (Sigma-Aldrich, St Louis, MO, USA) administration (4 mg per ml in the drinking water, about 300 mg/kg/d [29]) throughout gestation and lactation, and the control group (CH-CT, n = 8 and HF-CT, n = 8) supplied with normal drinking water. Dams were weighed weekly during gestation and lactation. The day of parturition is recorded as postnatal day (PND) 0. On PND1, the litter size was culled to ten (♂:♀ = 1:1) per litter. Pups’ body weight (BW) was monitored weekly. All pups were given CHOW diet after weaning. The offspring were weighed weekly (Figure 1).

### 2.2. Tissue Collection

The animals were fasted and water deprivation for four hours before sacrifice. At weaning, one male and one female offspring per litter and all the dams were sacrificed. At 18 weeks of age, another one male and one female offspring per litter were decapitated. Plasma of offspring and dams was collected for blood profile analyses. The gastrocnemius of offspring was quickly removed and divided into three parts: one part was snap-frozen and stored at −80 °C for subsequent molecular analyses; one part was fixed in 4% paraformaldehyde for hematoxylin and eosin (H&E) staining; the other part was fixed in 2.5% glutaraldehyde for transmission electron microscopy analysis.

### 2.3. Blood Profile Analyses

Blood glucose was determined by OneTouch Ultra 2 (LifeScan, Milpitas, CA, USA) via a small tail nick before sacrifice. Plasma insulin, leptin, triglyceride and cholesterol concentrations were determined by ELISA kits (Meimian, Jiangsu, China) in strict conformity with the manufacturer’s instructions. 

### 2.4. Quantitative Real-Time PCR Analysis

Quantitative real-time PCR (qPCR) assay was used to detect relative mRNA expression of genes related to the myogenesis and mitochondrial biogenesis and dynamics. The MRFs family include: MyoD, Myog, Myf5 and MRF4. The mitochondrial biogenesis-related genes include: Ppargc1a, Tfam, Nrf1. The mitochondrial dynamics-related genes include: Opa1, Drp1, Mfn1, Mfn2. Total RNA was isolated from muscle homogenates using animal RNA isolation kit (R0027, Beyotime Biotechnology, Beijing, China). RNA concentration and purity was measured using NuDrop (ACTGene, Piscataway, NJ, USA). Reverse transcription kit (K1622, Thermo Scientific, Waltham, MA, USA) was used to reverse transcribe RNA into cDNA. Gene expression was determined by qPCR using SYBR green dye with gene specific primer sets and an iQ5 PCR thermal cycler (Bio-Rad, Hercules, CA, USA) with the following cycle parameters: 95 °C for 3 min; 40 cycles for 95 °C for 10 s, 60 °C for 30 s and 72 °C for 30 s. Melting curves were performed to prove the uniqueness of products. The primers of the genes studied are listed in Table 1. The ^−ΔΔ^Ct method was used to determine relative expression values. Duplicate cross threshold values for each sample were averaged and subtracted from those derived from the housekeeping gene Actb.

### 2.5. Western Blotting

Total gastrocnemius protein was extracted using a total protein extraction kit for animal tissues (Invent Biotechnologies, Inc., Plymouth, MN, USA), with protease and phosphatase inhibitor (Roche, Mannheim, Germany). Total protein concentration was determined using a Pierce Rapid Gold BCA protein assay kit (Thermo Scientific, Waltham, MA, USA) according to the manufacturer’s instructions. Detailed method of Western blotting has been described previously [30], including imaging and data analysis. Antibody (Cell Signaling Technology, Danvers, MA, USA) details were as follows: total and phospho-AMPK at Thr172 (5831/2531), total and phospho-S6 ribosomal protein (rpS6) at Ser235/236 (2217/4858), total and phosphorylated eukaryotic translation initiation factor 4E-binding protein 1 (4E-BP1) at Thr37/46 (9644/2855).

### 2.6. Gastrocnemius Histology

The gastrocnemius was fixed in 4% paraformaldehyde solution, paraffin-embedded and cut into 3 µm thick sections. The paraffin sections were then stained with H&E and observed by light microscopy with digital camera (OLYMPUS, Tokyo, Japan). The muscle cross-sectional area, relative density of gastrocnemius and the percentage of interstitium were evaluated in ten different microscopic fields at 400× magnification of each muscle sample and quantified using the ImageJ 1.49v software (National Institutes of Health, Bethesda, MD, USA).

### 2.7. Transmission Electron Microscopy

After being isolated, gastrocnemius was fixed with 2.5% glutaraldehyde (Solarbio, Beijing, China) and then cut into blocks the size of one square millimeter using a scalpel. The preparation method of transmission electron microscope sample was previously described by Glauce et al. [31]. The mitochondrial morphology was examined by a JEOL 1010 (Akashina, Japan) transmission electron microscope and quantified using the ImageJ 1.49v software (National Institutes of Health, Bethesda, MD, USA). Systematic uniform random sampling (thirty digital images chosen for each group) was performed at a magnification of 30,000×. The number-weighted mean volume of mitochondria was obtained by dividing the total mitochondrial volume by the total number of mitochondria.

### 2.8. Statistical Analysis

Statistical analyses were performed using Prism 9 (GraphPad Software, San Diego, CA, USA). All data are expressed as mean ± SEM and were analyzed by a two-way ANOVA with factors of diet (CHOW, HF) and metformin (control, MET). One male and one female offspring from each litter were used as one individual at each time point in statistical analysis. Results were considered significant when *p* < 0.05.

## 3. Results

### 3.1. Phenotypes of Dams

The phenotypes of dams are shown in Table 2. There was no difference in maternal BW before pregnancy. CH-MT and HF-MT dams weighed significantly less than CH-CT and HF-CT dams both on gestation day (GD) 10 (main effect of metformin, *p* < 0.05) and GD20 (main effect of metformin, *p* < 0.05). On PND21, CHOW-fed dams had greater BW than HF-fed dams (main effect of HF diet, *p* < 0.05). There were no significant differences in dams’ blood glucose among the four groups at weaning. Maternal HF diet increased plasma insulin levels in dams (main effect of HF diet, *p* < 0.05), while metformin administration decreased plasma insulin levels (main effect of metformin, *p* < 0.05). Plasma triglyceride (main effect of metformin, *p* < 0.05) and cholesterol (main effect of metformin, *p* < 0.05) levels in MT groups were lower than CT groups. Four groups did not show significant differences in plasma leptin levels.

### 3.2. Phenotypes of Offspring

At postnatal week 3 and 6, male offspring in maternal HF groups had greater BW than that in maternal CT groups (main effect of maternal HF diet, *p* < 0.05, Figure 2A). At week 18, maternal HF diet increased male offspring’s BW (main effect of maternal HF diet, *p* < 0.05), while maternal metformin administration decreased male offspring’s BW (main effect of maternal metformin, *p* < 0.05) (Figure 2A). For female offspring, there was a significant main effect of maternal HF diet in increasing BW at postnatal week 3, 6 and 9 (*p* < 0.05, Figure 2B), while there was no difference in BW at week 18. Notably, in PND21 male, 18W male and 18W female, maternal HF diet led to elevated plasma insulin levels (main effect of maternal HF diet, *p* < 0.05), whereas maternal metformin administration reduced insulin levels (main effect of maternal metformin, *p* < 0.05) (Figure 2C,E). As for PND21 female offspring, maternal metformin administration decreased plasma insulin levels (main effect of maternal metformin, *p* < 0.05, Figure 2E), while there was no effect of maternal HF diet on plasma insulin levels. For PND21 and 18W male/female offspring, maternal metformin administration could decrease plasma leptin levels (main effect of maternal metformin, *p* < 0.05, Figure 2D,F). In addition, there was an interaction effect of maternal HF diet and metformin on plasma leptin levels in adult male offspring (*p* < 0.05, Figure 2D). Maternal metformin administration reduced triglyceride and cholesterol levels in both male and female offspring on PND21 (main effect of maternal metformin, *p* < 0.05), while the effect of maternal metformin disappeared in adult offspring (Figure 2G–J). Furthermore, maternal HF diet elevated plasma triglyceride levels in female offspring on PND21 (main effect of maternal HF diet, *p* < 0.05, Figure 2I).

### 3.3. Myogenesis Gene Expression and Morphology in SM of Offspring

We analyzed all four genes in the myogenic regulatory factor family including MyoD, Myog, Myf5 and MRF4 within SM in offspring on PND21 and in adulthood. In PND21 male offspring, maternal HF diet decreased mRNA expression of MRF4 (main effect of maternal HF diet, *p* < 0.05), while maternal metformin increased MRF4 expression (main effect of maternal metformin, *p* < 0.05, Figure 3A). There were no differences in gene expression among the four groups in PND21 female offspring (Figure 3B). In adult male offspring, maternal HF diet decreased mRNA expression of Myf5 and Myog (main effect of maternal HF diet, *p* < 0.05), while maternal metformin increased mRNA expression of Myf5 and MyoD (main effect of maternal metformin, *p* < 0.05, Figure 3C). Furthermore, there was an interaction effect of maternal HF diet and metformin on Myog expression in adult male offspring (*p* < 0.05, Figure 3C). In adult female offspring, maternal HF diet decreased mRNA expression of MyoD and Myog (main effect of maternal HF diet, *p* < 0.05), while maternal metformin increased mRNA expression of MyoD and MRF4 (main effect of maternal metformin, *p* < 0.05, Figure 3D).

To show the condition of muscle growth more intuitively, H&E staining was performed on the SM. In the HF-CT group (PND21/adult male and female offspring), we observed myocyte nucleus migration from the edge to the center of muscle fiber (yellow arrow, Figure 3E), a sign of muscle dysfunction [32]. Compared with maternal CHOW groups, maternal HF diet increased the percentage of interstitium in PND21/adult male and female offspring (main effect of maternal HF diet, *p* < 0.05, Figure 3F,G). No significant differences in the cross-sectional area of SM were found among the four groups (Figure 3H,I).

### 3.4. Mitochondrial Biogenesis and Dynamics Gene Expression in SM of Offspring

To investigate the role of maternal HF diet and metformin during pregnancy and lactation in mitochondrial biogenesis in the SM of offspring, we determined mRNA levels of genes regulating mitochondrial biogenesis: Ppargc1a, Tfam and Nrf1. The effect of maternal metformin decreasing Ppargc1a expression in both male and female offspring only occurred in adulthood (main effect of maternal metformin, *p* < 0.05, Figure 4C,D), but not in PND21 offspring (Figure 4A,B). Maternal HF diet increased mRNA expression of Tfam in PND21 male offspring (main effect of maternal HF diet, *p* < 0.05, Figure 4A), while there was a main effect of maternal metformin on Tfam expression in adult male offspring (main effect of maternal metformin, *p* < 0.05, Figure 4C). There were no differences in Nrf1 expression among the four groups in PND21/adult male or female offspring (Figure 4A–D).

Mitochondria are dynamic organelles which can change their morphology by fusion and fission. We measured mRNA levels of genes regulating mitochondrial dynamics: Opa1, Drp1, Mfn1 and Mfn2. Regardless of age, the effects of maternal metformin on Opa1 expression were found only in female offspring (main effect of maternal metformin, *p* < 0.05, Figure 4B,D), and there was an interaction effect of maternal HF diet and metformin on Opa1 expression in adult female offspring (*p* < 0.05, Figure 4D). Maternal HF diet increased mRNA expression of Drp1 in adult male offspring (main effect of maternal HF diet, *p* < 0.05, Figure 4C), while maternal metformin decreased Drp1 expression in adult male and female offspring (main effect of maternal metformin, *p* < 0.05, Figure 4C,D). Maternal metformin decreased mRNA expression of Mfn1 only in adult female offspring (main effect of maternal metformin, *p* < 0.05, Figure 4D). Maternal HF diet increased mRNA expression of Mfn2 in PND21 female and adult male/female offspring (main effect of maternal HF diet, *p* < 0.05, Figure 4B–D), while maternal metformin just decreased Mfn2 expression in adult males and females (main effect of maternal metformin, *p* < 0.05, Figure 4C,D). There were no differences in Drp1 or Mfn1 expression among the four groups in PND21 male or female offspring (Figure 4A,B).

### 3.5. Mitochondrial Number and Average Volume in SM of Offspring

Transmission electron micrographs displayed regular myofilament alignment and mitochondria from SM in CH-CT, CH-MT and HF-MT groups of PND21 male offspring. Maternal HF diet resulted in myofibrillar variations and mitochondrial swelling and vacuolization (highlighted in red) in HF-CT group (Figure 5A). Increase in mitochondrial number (Figure 5B) and average volume per vision (28.2 μm^2^) (Figure 5C) was observed in the MET groups as compared to controls (main effect of maternal metformin, *p* < 0.05).

### 3.6. AMPK and mTOR Signaling in SM of Offspring

Metformin has been shown to improve mitochondrial respiratory activity by increasing the phosphorylation level of AMPK at T172 [19]. AMPK and mTOR are sensors of cellular energy and nutrient levels, and acute modification of the energy metabolism system can be achieved through phosphorylation [3]. We detected the phosphorylation at T172 of AMPK and found that maternal metformin could increase it regardless of age and gender (main effect of maternal metformin, *p* < 0.05, Figure 6B,F). Furthermore, there was a main effect of maternal HF diet on decreasing AMPK phosphorylation (*p* < 0.05) and an interaction effect of maternal HF diet and metformin on AMPK phosphorylation (*p* < 0.05) in 18W female offspring (Figure 6F). rpS6 and 4E-BP1 are two downstream molecules of mTORC1 pathway. The phosphorylation of rpS6 was not affected by maternal HF diet or metformin (Figure 6C,G). In PND21 male offspring, maternal metformin reduced phosphorylation of 4E-BP1 (main effect of maternal metformin, *p* < 0.05), and there was an interaction effect of maternal HF diet and metformin on 4E-BP1 phosphorylation (*p* < 0.05, Figure 6D). At 18W, there was both a main effect of maternal HF diet (*p* < 0.05) and a main effect of maternal metformin (*p* < 0.05) on 4E-BP1 phosphorylation in male and female offspring (Figure 6D,H).

## 4. Discussion

Both maternal diet and metformin administration during gestation and lactation influence the development and metabolism of offspring’s SM. We demonstrated that maternal metformin exposure during gestation reduced maternal body weight over the course of pregnancy. On PND21, the reason why CHOW-fed dams had greater BW than HF-fed dams may be that offspring with maternal HF diet consume more breast milk [33], which causes higher energy expenditure in HF-fed dams. This may be related to increased proportion of C18:2 free fatty acid and fatty acyl residues in the milk of HF-fed dams [34] and hyperphagic offspring of HF-fed rat dams [35]. Consistent with our previous study [36], exposure to HF diet during gestation and lactation period is sufficient to predispose the offspring to metabolic disorders. Our study shows that maternal metformin decreased the plasma levels of insulin, triglyceride and cholesterol in both dams and offspring. Metformin is also known to reduce fasting insulin and BW in HF-fed rats, primarily by improving HF diet induced muscle insulin resistance [37]. In the study of Geerling et al., metformin reduced plasma triglycerides by promoting the clearance of VLDL-triglycerides by brown adipose tissue in mice [38].

According to previous studies, MyoD and myogenin were inhibited by fatty acids-, diacylglycerols- and ceramides-induced apoptosis [39]. A local proinflammatory status caused by adipose tissue from obese subject increased SM inflammation, which can attenuate SM myogenesis [40]. Maternal metformin decreased obesogenic diet-induced fatty acid changes and inflammation in liver of offspring [41]. Thus, we speculate that maternal metformin increased offspring SM myogenesis due to reduced fatty acid and inflammation in this study. Interestingly, we found that maternal metformin is more likely to cause genetic changes in adulthood rather than PND21 in both males and females. As Lu etc. demonstrated, pregnant mice demonstrated accelerating muscle healing, and muscle progenitor cells (MPCs) from non-pregnant mice showed enhanced myogenic ability when cultured in the presence of pregnant mouse serum [42]. In view of this, we speculated that PND21 offspring may obtain enough myogenic ability because of the MPCs obtained from their mothers [43]. Thus, there was less difference in MRFs expression of SM on PND21. Consistent with our results, maternal HF diet led to offspring SM remodeling, with excessive extramyocyte space [44], which may finally cause SM dysfunction, including insulin resistance [45]. The present data suggest that maternal metformin could improve decreased myogenesis, muscle dysfunction and increased extramyocyte space caused by maternal HF diet in the offspring.

Increasing evidence suggests that metformin induces extrahepatic tissue-specific effects, for example, mitochondria are considered as the main cellular targets in SM [46]. The impairment of mitochondrial biogenesis is associated with metabolic diseases, such as T2D and obesity [47]. A high level of reactive oxygen species (ROS) production in skeletal muscle caused by hyperlipidemia is associated with mitochondrial dysfunction, which induced mitochondrial biogenesis [48]. Consistently, our results show that maternal HF diet increased mitochondrial biogenesis-related gene (Tfam) in PND21 male offspring. Interestingly, we found that maternal metformin changed mRNA levels of Ppargc1a only in adult offspring, while Tfam was changed both at weaning and in adulthood. As for Nrf1, there was no significant difference among each group. Based on research of Xia et al. [49] and Dorn et al. [50], Nrf1 precedes changes in Tfam, so we speculate that the change of Nrf1 may occur earlier than PND21. Exposure to maternal HF diet can cause life-long mitochondrial changes in offspring SM [51]. We speculate that maternal HF diet-caused cellular stress response could be reversed by maternal metformin through regulating the form, volume and function of mitochondria to maintain mitochondrial homeostasis. Unexpectedly, maternal metformin reduced the expression of mitochondrial biogenesis- and dynamics-related genes, but increased the number of mitochondria in SM of offspring, which is worthy of further study.

AMPK, as a key factor for cell metabolism to adapt to glucose levels, is an efficient drug target in the therapeutic applications in T2D and obesity [52]. Metformin-induced AMPK activation may indirectly reduce gluconeogenesis by increasing hepatic insulin sensitivity [17], so as to improve the increased insulin levels caused by maternal HF diet in this study. Damaged mitochondria, caused by HF diet-induced overproduction of ROS in mitochondria, can be degraded by lysosomal, while mitochondria could maintain homeostasis through undergoing constant mitochondrial fission and fusion [53]. Metformin could attenuate ROS via AMPK activation [54], so that it could protect mitochondria from damage. Francesca et al. demonstrated that maternal overweight activated mTORC1, possibly through the inhibition of AMPK in rat placenta [55]. Metformin could promote the dephosphorylation in p-4E-BP1 by inhibiting mTORC1 [23]. In addition, according to Li’s research, the inhibition of mTOR pathway can increase myogenesis [56], suggesting that maternal metformin could increase myogenesis through inhibiting mTORC1 in this study. A previous study showed that inhibition of mTOR could decrease the gene expression of the mitochondrial transcriptional regulators Ppargc1a in SM, resulting in a decrease in mitochondrial gene expression [21], which is consistent with the effect of maternal metformin in our study.

So far, many studies have explored gender differences related to obesity, including sex hormones [57], adipokine signaling pathway [58] and appetite regulation [59]. Males are more likely to become obese than females because ovarian hormones may protect females from disturbance [60]. In our study, there was no effect of maternal HF diet or metformin on adult female offspring’s body weight, which is consistent with the previous studies. Androgens, to some extent, are thought to contribute to anabolic actions of muscle [61], which may be the main reason for gender differences in myogenesis in our study.

There are a few limitations in this study: metformin can easily cross the placenta during gestation [62], and offspring could contact metformin water bottles during late lactation. We cannot rule out the possibility that offspring have direct exposure to metformin during the fetal period or late lactation period. We are interested in the metformin concentration in milk or in pups’ circulation during lactation, and ^11^C-metformin PET may help us to verify it [63]. Notably, it is challenging if we try to distinguish these two mechanisms clearly in this experimental setting.

Controversies persist over whether to use metformin in the treatment of GDM due to the lack of clinical data on fetal development [15]. Our data show that maternal HF diet during pregnancy and lactation has negative effects on the overall development of offspring, while maternal metformin administration during pregnancy and lactation improves the growth and development of offspring SM of HF-fed mothers through AMPK/mTOR pathways. Our data provide positive evidence for the use of metformin in pregnant women. The effect of maternal metformin use on other target organs needs further investigation.

## Figures and Tables

**Figure 1 nutrients-13-03417-f001:**
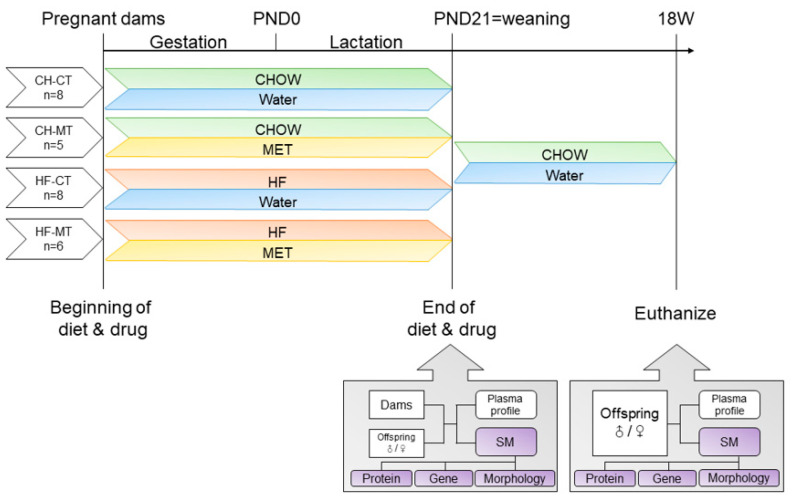
Schematic illustration of experimental groups. Dams were fed CHOW or high fat (HF) diet and given water with or without metformin (MET) throughout gestation and lactation. After weaning, offspring were all fed CHOW diet and given water without metformin. Plasma and skeletal muscle (SM) samples were collected on postnatal day (PND) 21 and at 18 weeks separately.

**Figure 2 nutrients-13-03417-f002:**
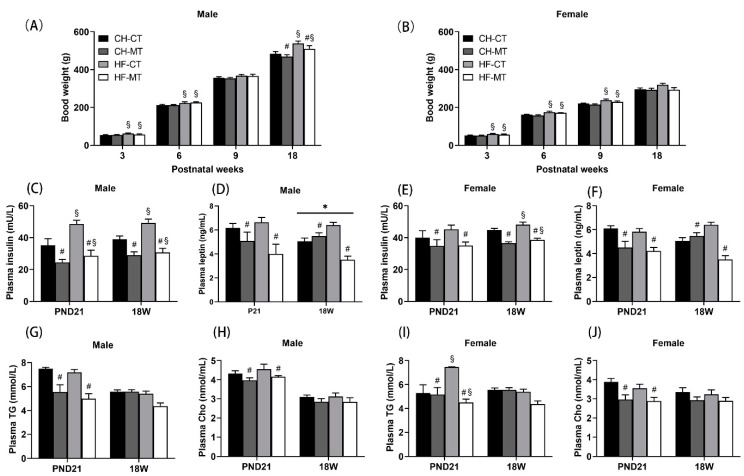
Body weight and plasma profile of offspring. Male (**A**) and female (**B**) offspring were weighed once a week. Plasma insulin (**C**), leptin (**D**), triglyceride (TG) (**G**) and total cholesterol (Cho) (**H**) of male offspring were measured at weaning and 18 weeks. Plasma insulin (**E**), leptin (**F**), TG (**I**) and Cho (**J**) of female offspring were measured at weaning and 18 weeks. Data are expressed as mean ± S.E.M (n = 5–8 per group). Data were analyzed by two-way ANOVA, ^#^ main effect of maternal metformin, *p* < 0.05 versus control, ^§^ main effect of maternal HF diet, *p* < 0.05 versus maternal CHOW diet, * interaction effect of maternal HF diet and metformin, *p* < 0.05.

**Figure 3 nutrients-13-03417-f003:**
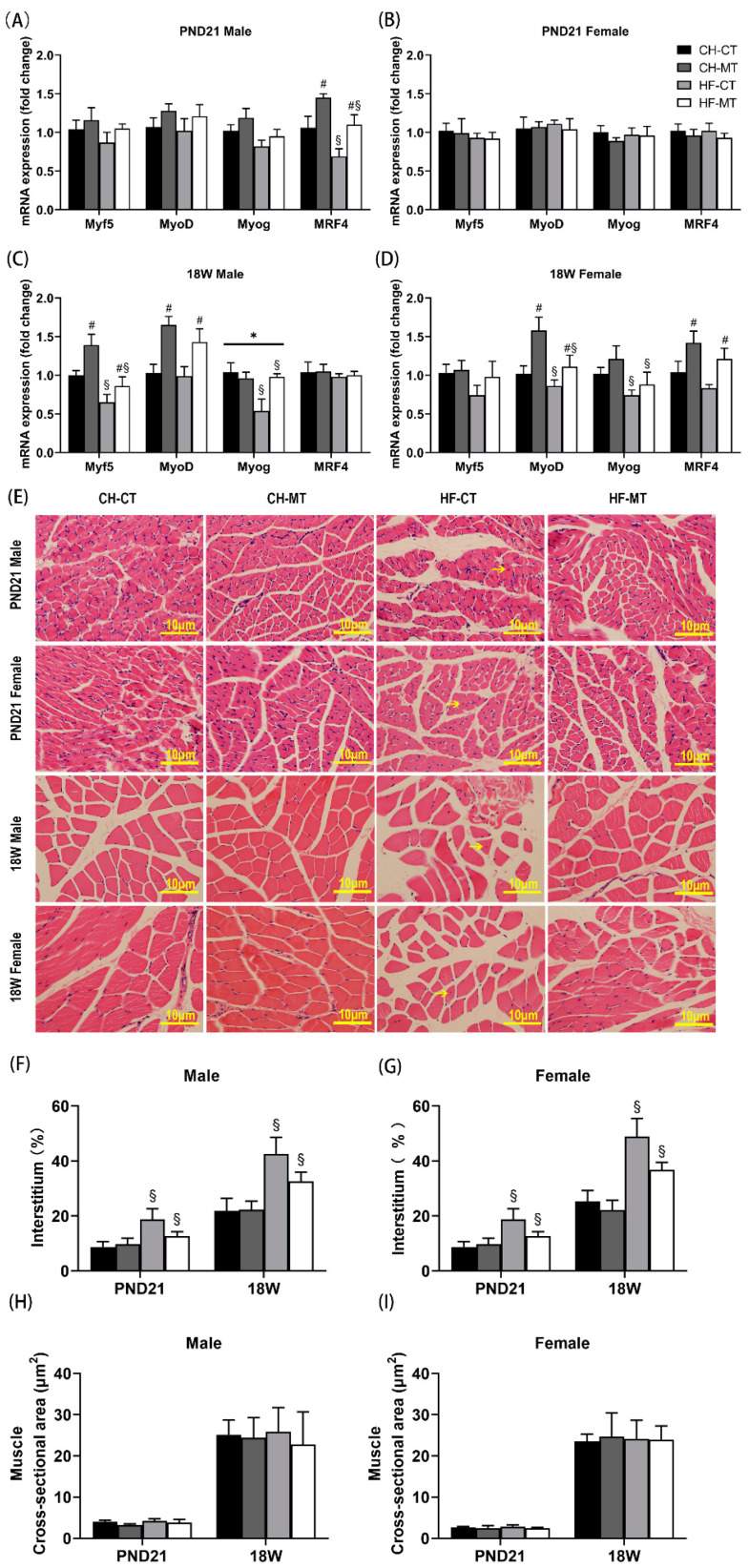
mRNA expression of myogenesis-related genes and representative histological sections of offspring skeletal muscle. mRNA expression of myogenesis-related genes was measured in skeletal muscle of PND21 males (**A**), PND21 females (**B**), 18W males (**C**) and 18W females (**D**). (**E**) H&E staining of representative histological sections of skeletal muscle. Scale bar = 10 μm. Percentage of interstitium in muscle of male (**F**) and female (**G**) offspring, and muscle cross-sectional area of male (**H**) and female (**I**) offspring were analyzed. Data are expressed as mean ± S.E.M (n = 5–8 per group). Data were analyzed by two-way ANOVA, ^#^ main effect of maternal metformin, *p* < 0.05 versus control, ^§^ main effect of maternal HF diet, *p* < 0.05 versus maternal CHOW diet, * interaction effect of maternal HF diet and metformin, *p* < 0.05. Myf5, myogenic factor 5; MyoD, myogenic differentiation antigen; Myog, myogenin; MRF4, muscle regulatory factor 4.

**Figure 4 nutrients-13-03417-f004:**
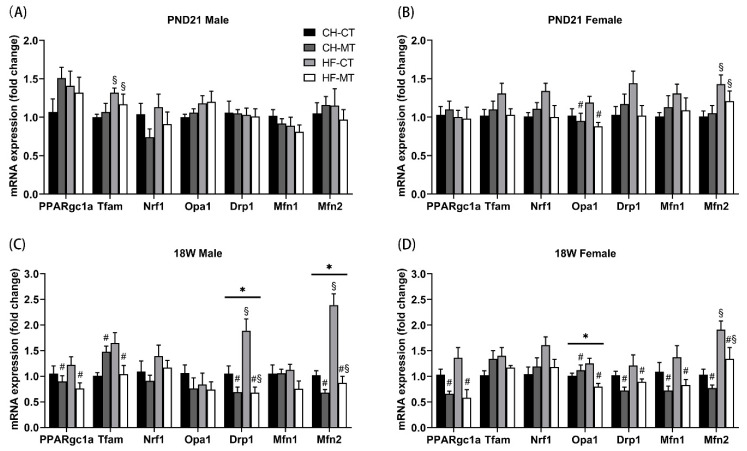
mRNA expression of mitochondrial biogenesis- and dynamics-related genes in skeletal muscle of offspring. mRNA expression of mitochondrial biogenesis- and dynamics-related genes in skeletal muscle was measured in PND21 males (**A**), PND21 females (**B**), 18W males (**C**) and 18W females (**D**). Data are expressed as mean ± S.E.M (n = 5–8 per group). Data were analyzed by two-way ANOVA, ^#^ main effect of maternal metformin, *p* < 0.05 versus control, ^§^ main effect of maternal HF diet, *p* < 0.05 versus maternal CHOW diet, * interaction effect of maternal HF diet and metformin, *p* < 0.05. Drp1, dynamic-related protein 1; Mfn, dynamin-related GTPase termed mitofusin; Nrf, nuclear respiratory factor; Opa1, Optic atrophy protein 1; Ppargc1a, peroxisome proliferator-activated receptor gamma coactivator 1-alpha; Tfam, mitochondrial transcription factor A.

**Figure 5 nutrients-13-03417-f005:**
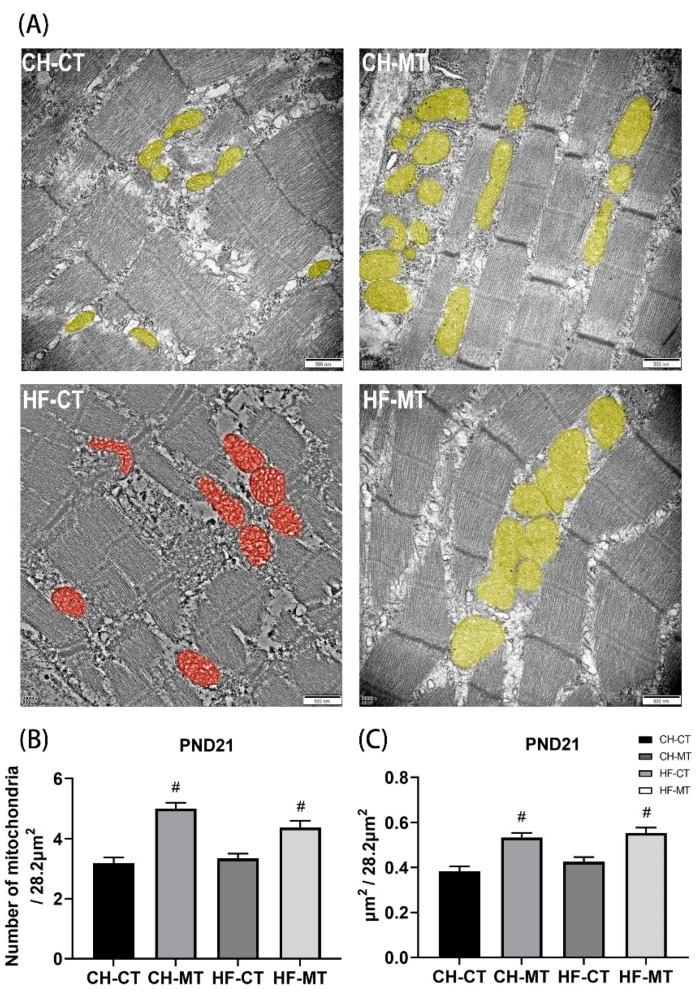
Transmission electron micrographs of skeletal muscle in PND21 male offspring. (**A**) Representative transmission electron micrographs. Mitochondria were artificially colored. Mitochondrial number (**B**) and average volume (**C**) per vision (28.2 μm^2^) were measured. Scale bar = 500 nm. Data are expressed as mean ± S.E.M (n = 4–5 per group). Data were analyzed by two-way ANOVA, ^#^ main effect of maternal metformin, *p* < 0.05 versus control.

**Figure 6 nutrients-13-03417-f006:**
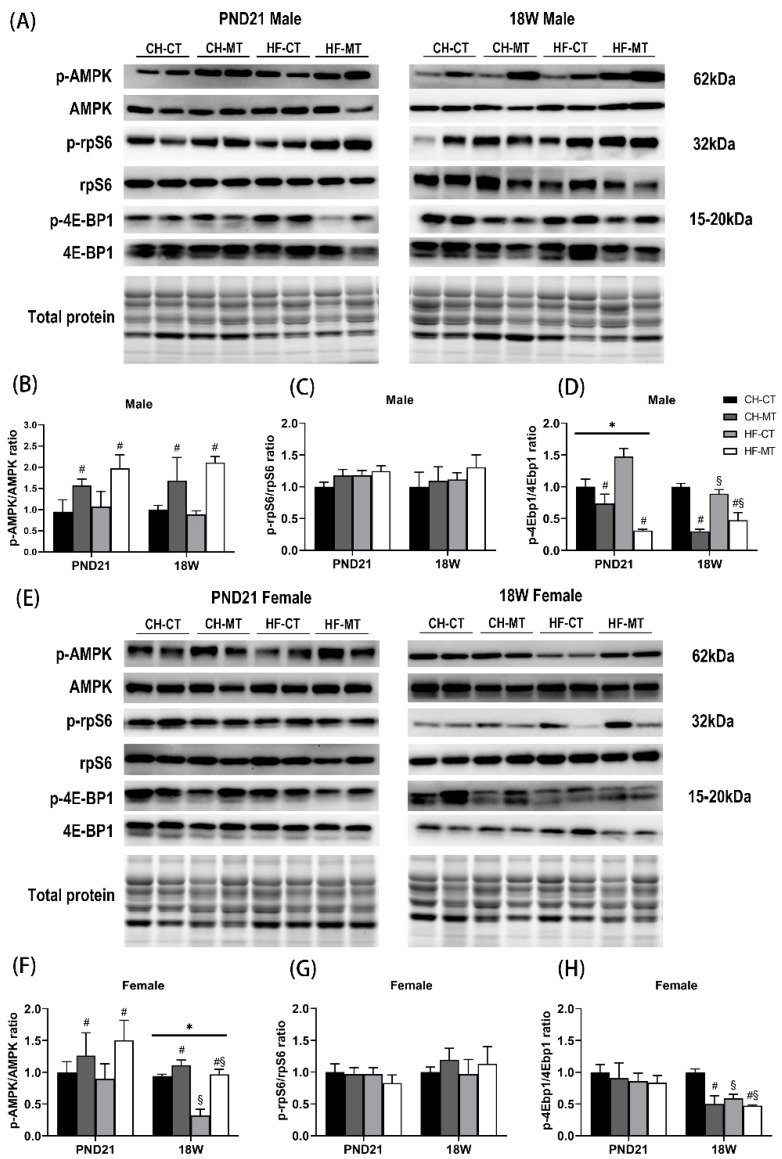
Expression of total and phosphorylated AMP-activated protein kinase (AMPK), ribosomal protein S6 (rpS6) and eukaryotic initiation factor 4E-binding protein 1 (4E-BP1) in skeletal muscle of offspring. (**A**) Representative Western blots of total and phosphorylated AMPK, rpS6 and 4E-BP1 in skeletal muscle of PND21 and 18W male offspring. Histograms summarized phosphorylated-to-total ratio of AMPK (**B**), rpS6 (**C**) and 4E-BP1 (**D**) in male offspring. (**E**) Representative Western blots of total and phosphorylated AMPK, rpS6 and 4E-BP1 in skeletal muscle of PND21 and 18W female offspring. Histograms summarized phosphorylated-to-total ratio of AMPK (**F**), rpS6 (**G**) and 4E-BP1 (**H**) in female offspring. Data are expressed as mean ± S.E.M (n = 5–8 per group). Data were analyzed by two-way ANOVA, ^#^ main effect of maternal metformin, *p* < 0.05 versus control, ^§^ main effect of maternal HF diet, *p* < 0.05 versus maternal CHOW diet, * interaction effect of maternal HF diet and metformin, *p* < 0.05.

**Table 1 nutrients-13-03417-t001:** Summary of qPCR oligonucleotide primers used in measuring mRNA expression of genes related to myogenesis, mitochondrial biogenesis and dynamics.

Gene	Gene Bank No.	DNA Sequence (5′-3′)
Myogenesis		
MyoD	NM_176079.1	Forward Primer GACGGCTCTCTCTGCTCC Reverse Primer AAGTGTGCGTGCTCCTCC
Myog	NM_017115.2	Forward Primer GAGCGCGATCTCCGCTCAAGAG Reverse Primer CTGGCTTGTGGCAGCCCAGG
Myf5	NM_001106783.1	Forward Primer GAGCCAAGAGTAGCAGCCTTCG Reverse Primer GTTCTTTCGGGACCAGACAGGG
MRF4	NM_013172.2	Forward Primer AGAGACTGCCCAAGGTGGAGATTC Reverse Primer AAGACTGCTGGAGGCTGAGGCATC
Mitochondrial biogenesis		
Ppargc1a	NM_031347.1	Forward Primer GACACGAGGAAAGGAAGACTAAA Reverse Primer GTCTTGGAGCTCCTGTGATATG
Tfam	NM_031326.1	Forward Primer CTGATGGGCTTAGAGAAGGAAG Reverse Primer GTTATATGCTGACCGAGGTCTTT
Nrf1	NM_001100708.1	Forward Primer GCTCATCCAGGTTGGTACTG Reverse Primer CCATCAGCCACAGCAGAATA
Mitochondrial dynamics		
Opa1	NM_133585.3	Forward Primer GAGTATCAAGCGGCACAAATG Reverse Primer CGTCCCACTGTTGCTTATCT
Drp1	NM_053655.3	Forward Primer TGTGGTGGTGCTAGGATTTG Reverse Primer TGGCGGTCAAGATGTCAATAG
Mfn1	NM_138976.1	Forward Primer AACAGCACACTATCAGAGCTAAA Reverse Primer GATTTGGTCTTCCCTCTCTTCC
Mfn2	NM_130894.4	Forward Primer CAGTGTTTCTCCCTCAGCTATG Reverse Primer TAGGGCCCAGGAACCTATT
Housekeeping genes		
Actb	NM_031144.3	Forward Primer ACAGGATGCAGAAGGAGATTAC Reverse Primer ACAGTGAGGCCAGGATAGA

Actb, beta actin; Drp1, dynamic-related protein 1; Mfn, dynamin-related GTPase termed mitofusin; MyoD, myogenic differentiation antigen; Myog, myogenin; Myf5, myogenic factor 5; MRF4, muscle regulatory factor 4; Nrf, nuclear respiratory factor; Opa1, Optic atrophy protein 1; Ppargc1a, peroxisome proliferator-activated receptor gamma coactivator 1-alpha; Tfam, mitochondrial transcription factor A.

**Table 2 nutrients-13-03417-t002:** Maternal body weight during gestation and lactation and plasma profile at weaning.

	CH-CT (n = 8)	CH-MT (n = 5)	HF-CT (n = 8)	HF-MT (n = 6)
Body weight (g)				
Before pregnancy	209.1 ± 5.1	209.7 ± 5.1	211.4 ± 4.6	207.5 ± 6.5
GD10	265.4 ± 7.9	201.4 ± 16.9 ^#^	265.9 ± 6.1	234.5 ± 6.4 ^#^
GD20	342.5 ± 15.8	298.0 ± 14.4 ^#^	337.0 ± 8.7	284.7 ± 6.7 ^#^
PND21	296.9 ± 6.4	293.8 ± 8.9	270.4 ± 6.4 ^§^	262.5 ± 6.7 ^§^
Plasma profile				
Glucose (mmol L^−1^)	8.6 ± 0.3	8.7 ± 0.1	8.0 ± 0.2	8.7 ± 0.2
Insulin (mU L^−1^)	35.9 ± 1.3	26.7 ± 1.7 ^#^	43.7 ± 2.4 ^§^	30.8 ± 2.7 ^#,§^
TG (mmol L^−1^)	5.3 ± 0.3	4.5 ± 0.5 ^#^	6.4 ± 0.6	4.3 ± 0.5 ^#^
Cho (nmol mL^−1^)	2.4 ± 0.2	2.1 ± 0.1 ^#^	2.8 ± 0.2	2.1 ± 0.2 ^#^
Leptin (ng mL^−1^)	5.2 ± 0.7	5.3 ± 0.5	5.1 ± 0.6	5.2 ± 0.6

Values are expressed as mean ± standard error of mean (SEM). n, number of dams per group; GD, gestation day; PND, postnatal day; TG, triglyceride; Cho, Cholesterol. Data were analyzed by a two-way ANOVA, ^#^ main effect of maternal metformin, *p* < 0.05 versus control, ^§^ main effect of maternal HF diet, *p* < 0.05 versus maternal CHOW diet.

## Data Availability

The data presented in this study are available on request from the corresponding author.

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
