# Peer review of "Maternal Metformin Treatment during Gestation and Lactation Improves Skeletal Muscle Development in Offspring of Rat Dams Fed High-Fat Diet"

_nutrients, 2021, doi:10.3390/nu13103417_

Round 1

Reviewer 1 Report

Manuscript nutrients-1374620
Review
Scope of the journal: I think the paper is in the scope of Nutritiens, but as a clinician I found it more suitable in a preclinical journal.
Importance: This study is of great importance since it studies gestational diabetes mellitus, which is an emerging problem world wide.
The study design is very good. It is an animal study, randomized to four different groups.
Soundness of conclusions and interpretation are in perfect order.
The relevance of the discussion is great and very relevant.
The paper is very clear written.
Page 2, line 46. …SM is a potential anti-obesity since it target around 40% of the body. Do the authors mean human body? Is the reference 3 agequate?
Page 2, line 70. AMPK, please write the abbreviation when it is used for the first time?
Page 2, Animals. Was the study blinded in any way?

Tables and figures are relevant
References up to date and relevant.

Reviewer 2 Report

In the present experimental design, the administration of metformin was carried out in the dams, not directly in the offspring. However, the effects on the offspring are clear and well presented. However, it is difficult to attribute to direct or indirect effects of metformin. The authors discuss this issue well as a study limitation at lines 431-435 of the discussion. However, in order to clarify this effect, is milk available to measure metformin in milk? Could it be possible to measure circulating metformin levels in pups at least in PN21?

Figure 3. Correct "exprssion"
